# Nurturing families: One year pilot outcomes for a modified Parent Child Assistance Program in Australia

Martyn Symons[1,2]*, Amy Finlay-Jones[1,2], Jennifer Meehan[3], Natalie Raymond[3], Rochelle Watkins[1,2]

1 National Health and Medical Research Council FASD Research Australia Centre of Research Excellence, West Perth, Australia, 2 Telethon Kids Institute, The University of Western Australia, Perth, Australia, 3 Women's Health and Family Services, Perth, Australia

* martyn.symons@telethonkids.org.au

**Data Availability Statement:** We are requesting that the data for this study are not made available due to their sensitive nature. The study is a pilot and only reports on outcomes for eight clients.

## Abstract

Alcohol and Other Drug (AOD) exposure during pregnancy is linked to serious adverse child outcomes, including Fetal Alcohol Spectrum Disorder (FASD). The Parent-Child Assistance Program (PCAP) supports women with problematic AOD use, who are pregnant or have young children, and are not effectively engaging with services. PCAP has been shown to reduce alcohol exposed pregnancies, promote AOD abstinence, increase employment and family planning and improve child outcomes. This manuscript reports the first pilot evaluation of the PCAP program delivered in Australia. A pre-post-intervention repeated measures design was used. Eleven women receiving PCAP from a not-for-profit organisation were invited to take part in the study, with eight providing complete pre-post data. Home visitation case management was provided by trained and experienced case-managers. Clients were assisted to engage with existing services effectively to meet their own goals via a combination of relational theory, motivational interviewing and harm reduction concepts. The PCAP Modified Addiction Severity Index 5th Edition was adapted for use in Australia and was used to measure domains of addiction severity related problems as the primary outcome. Secondary outcomes included client satisfaction and program fidelity. There were significant changes in composite addiction severity scores from baseline to one year. 80% of participants had periods of abstinence of longer than four months. All clients had better connection to services, no subsequent AOD exposed pregnancies, and were highly satisfied with the program. Four had children returned to their care. Implementation was similar to the original PCAP program with major differences including case-managers relying on training manuals only without undertaking in-person training; being more experienced; providing more direct AOD counselling; and having less supervision. The findings will inform future program delivery and methodology for a larger longitudinal study assessing outcomes at program exit.

These clients are part of a very vulnerable population and have experienced alcohol and drug use issues, contact with the justice system, domestic violence, and typically many adverse childhood experiences. The questionnaires used asked clients to provide information on issues with legal implications including illicit drug use and possible criminal behaviour including prostitution. Clients participated in the study with the understanding that the data would not be shared but only published in aggregate. Data requests can be made to the Head of the Data Strategy and Governance Unit at Telethon Kids Institute, Rad Aniba: Rad.Aniba@telethonkids.org.au.

**Funding:** This work was supported by an Australian National Health and Medical Research Council (NHMRC) Centre of Research Excellence grant (App1110341, MS, AFJ, RW). The provision of the Nurturing Families program has been made possible through funding provided by WA Primary Health Alliance through the Australian Government under the PHN Program to Women's Health and Family Services. The funders had no role in study design, data collection and analysis, decision to publish, or preparation of the manuscript.

**Competing interests:** The authors have declared that no competing interests exist.

## 1 Introduction

Alcohol and other drug use (AOD) during pregnancy is a serious public health issue in Australia. The 2019 National Drug Strategy Household Survey reported ~35% of women consumed alcohol during pregnancy, 10.8% smoked during pregnancy, 5.4% used cannabis and 1.8% illicit drugs before recognition of pregnancy [1]. Although alcohol use during pregnancy is falling [2], rates remain concerning given national guidelines recommend no alcohol use during pregnancy [3]. An unplanned pregnancy rate of around 40% in Australia [4] contributes to around 60% of women consuming alcohol between conception and recognition of their pregnancy [5].

Prenatal alcohol exposure (PAE) to the fetus is associated with several deleterious outcomes, including Fetal Alcohol Spectrum Disorder (FASD). FASD is a spectrum disorder characterised by severe impairments in at least three neurocognitive domains [6]. Individuals with FASD experience increased risk of physical and mental health comorbidities [7,8], as well as a range of adverse psychosocial consequences including educational and vocational difficulties and increased risk of contact with the justice system [9,10]. Together, these health and social costs are associated with an economic cost of approximately $23,000 USD per person per year with FASD [11]. The global prevalence of FASD has been estimated at 7.7 per 1000 [12].

Methamphetamine, cannabis and opioids also cross the placenta and are associated with lifespan neurodevelopmental and behavioural problems in offspring [13–15]. In Western Australia, as in other Australian states, methamphetamine exposure in pregnancy is an increasing concern [16]. Polysubstance use during pregnancy is frequent [17] and causes a synergistic increase in the risk of adverse outcomes for mothers and their children [18]. Addressing these issues requires responsiveness to social determinants of health and the impact of trauma and perinatal mental health difficulties on AOD behaviour [19–21]. The problem is often intergenerational and in a sample of 1400 women at high-risk of an AOD-exposed pregnancy, 87% had parents who used alcohol or drugs, 65% were abused as a child and 26% had experience in the foster care system [22].

Pregnancy is a good time to intervene as it provides a strong motivation for reducing substance use, and risk for drug abuse is substantially reduced through to the postpartum period [23]. However, engaging and retaining women in substance use interventions requires a trauma-focused and woman-centred approach that is responsive to the concerns and needs of the mother [16,21,24]. For example, Australian research has found that women who use methamphetamine in pregnancy report concerns that disclosing this may result in child removal [16,21]. Moreover, similar to substance abuse interventions for other target groups, risk of relapse following intervention for perinatal substance use is high [25]. For example, one study of pregnant women enrolled in a variety of substance use interventions found 83% of women achieved abstinence to at least one substance but 80% relapsed within two years, indicating a need for long-term support [17]. For treatment efficacy, it is important to address the wide-ranging needs of women in substance use treatment, as most have experienced multiple life traumas [20].

The Parent-Child Assistance Program (PCAP) program was developed at the University of Washington in the US [26] in response to poor treatment engagement and outcomes among mothers with intergenerational co-occurring addiction and psychiatric disorders. PCAP provides non-judgemental coaching and assistance over a sufficient time-period (three years) for women to make substantial and long-lasting behavioural changes. An evidence-based home visitation case-management model is provided to mothers with AOD use during pregnancy. The goal of the program is to help mothers achieve and maintain recovery, build healthy families, and prevent future alcohol and drug-exposed pregnancies.

The PCAP model is based on relational theory, motivational interviewing, and harm reduction principles. PCAP case managers develop supportive, empathic relationships with their clients, and conduct an average of two home visits per month, depending on the needs of the family. They have B.A. degrees in social service fields, average caseloads of 16 families, and are supervised by experienced clinicians. An essential role of PCAP case managers is to work with mothers to identify services needed, then coordinate and consult with appropriate providers to ensure that relevant services are received. Unlike many other AOD programs that withdraw help from clients who return to using AOD or have significant absences program, PCAP continues to provide support in these cases and case-managers ". . .never give up on a woman; women are never asked to leave the program because of relapse or setbacks." p.3 [27].

Studies in the US and Canada have consistently demonstrated the effectiveness of PCAP in assisting a majority of participants to access substance abuse treatment, reduce AOD use, maintain child custody, and prevent future births of AOD exposed infants [28–32]. Studies have also reported women in the program experiencing increased income [31], higher rates of children staying with their birth mother [31], reduction of alcohol exposed pregnancies [30], and increased use of family planning [29]. A Canadian economic evaluation for 366 women undertaking the PCAP program over three years estimated an incremental cost per prevented FASD case of CAD$97,000 and net monetary benefit of CAD $22 million [33]. This is backed up by Washington US data showing benefits including low rates of AOD exposed pregnancies, increased education, permanent housing and 80% of children living with their own families for over 1400 high-risk women at a cost of USD $5768 per client per year [22].

The Women's Health and Family Services (WHFS) Centre in Perth provides alcohol and drug counselling, mental health, domestic violence, Aboriginal family support, and perinatal care services. WHFS approached the authors to evaluate a pilot PCAP program in Perth that had initially received funding for one case-manager and during the pilot received additional funding for a second case-manager. This is the first pilot and evaluation of the PCAP program in Australia. The pilot project examined what modifications would be needed to adapt the PCAP program for Australia, tested the evaluation tools using an electronic data collection and integration system, and evaluated the potential efficacy of the program including satisfaction of clinical staff and clients. A key purpose for collecting this initial pilot data was for use in applying for further program funding.

## 2 Materials and methods

### 2.1 Ethical approval

Ethical approval was granted by The University of Western Australia Human Ethics Committee (RA/4/20/4289). Recruitment and participation followed Australian National Health and Medical Research Council guidelines with participants providing formal written consent. Due to the client vulnerability, it was emphasised that not taking part would not affect their position within the treatment program and they could withdraw at any time.

### 2.2 Participants

This size of this pilot was limited by the available budget which allowed one case-manager to work four days per week supporting 11 clients. All clients receiving the Nurturing Families modified PCAP program from WHFS in the Northern Metropolitan Health Region of Perth, Australia, were invited to participate. Case-managers introduced the research concept to clients and asked if they would be willing to meet with a researcher (MS) to undergo the formal recruitment process. One declined, one moved out of the area before consent and data collection, and one could not participate in the data collection window due to domestic violence.

Therefore, eight clients completed both baseline and follow-up interviews. Two clients identified as Australian, two as Australian Aboriginal, one Canadian, one Tongan and two were of unknown background. At baseline two clients were aged 25–29 years, three aged 30–34 years, and three aged 35–39 years. Seven clients lived in a residence not owned by them or their families. Four had not moved in the past year, the others had moved once, three, four and seven times. Four had dependents living with them. Education completed ranged from 9–15 years (15 indicating University Degree, mean = 12 years). Six were unemployed or busy with home duties, and two employed part-time. Five had never been married, one was in a de-facto relationship, one separated and one did not respond to this question. At follow-up one client was in remand in a secure facility.

## 2.3 Design

A pre-post intervention cohort design was used. Outcomes for clients were assessed at program entry (baseline) and again after approximately 12 months (mean = 12, range 7–24) of receiving the Nurturing Families intervention (follow-up) using the existing standard PCAP evaluation tool, the PCAP modified Addiction Severity Index 5th Edition [ASI, [34]]. Baseline measures were administered by the case-manager (JM) as part of typical clinical practice. Follow-up measures were collected by a researcher (MS) in the presence of the case-manager who provided emotional support to clients and assisted clients to recall specific details in areas such as AOD use and service use where clients forgot relevant details. The case-manager and researcher collaborated to ensure ASI responses were consistently recorded. Client satisfaction with the service was measured with the validated Treatment Perception Questionnaire [TPQ, [35]] by the researcher alone to ensure confidentiality. Clients debriefed with their case-manager after follow-up interviews to reduce potential trauma.

## 2.4 Measures

**2.4.1 The PCAP modified Addiction Severity Index, 5<sup>th</sup> Edition.** The ASI assesses AOD use, utilisation of alcohol/drug treatment, appropriate connection and access to community services, mental and physical health, family planning (use of birth control and subsequent pregnancies), income, and child health, wellbeing and custody. The ASI has been shown to be reliable and valid in a range of different settings conducted as an interview with treatment seeking populations [34]. The "Family History of Alcohol/Drug Psychiatric Problems" domain was introduced in the 5<sup>th</sup> edition and asks about the number and types of substance abuse and psychiatric problems in the patient's biological family [34]. The PCAP Modified ASI also has additional questions to evaluate program goals including the: T-ACE, a short scale for prenatal detection of risk-drinking [36]; negative experiences related to AOD use (e.g. relationship break-up, getting arrested, losing a job); childhood history including being adopted or fostered, school graduation and pregnancy, running away from home; the Adverse Childhood Experience Questionnaire [ACE, [37]]; questions about the status of the client's natural mother and her alcohol use during pregnancy; family planning, contraceptive use and number of biological children and count of children living with the client; the use of and connection with community services. An additional Part B collected information about the "Target Child" whom the client was pregnant with when enrolled or had most recently given birth to, including pregnancy outcome, AOD use during pregnancy, gestational age, demographics, who the child was discharged with, who has legal custody, child and protective services involvement, father's details, and doctor and midwife visits throughout pregnancy. At follow-up there were additional questions about: the target child including where they were living, time in custody, visits to doctors and child health nurses, immunisation status, accidents and illnesses;

subsequent pregnancies and births including AOD use and treatment; and the custody of other biological children.

Client history was collected in the ASI via: The ACE survey; previous disadvantage experienced by clients; and family history of AOD use using the questions "Have any of your relatives had what you would call a significant drinking, drug use or psychiatric problem—one that did or should have led to treatment?"

**2.4.2 Other ASI outcomes of interest.** Each section of the ASI concluded with questions rating how troubled or bothered clients have been in the past 30 days and how important it was to clients to get treatment for issues in that section. Clients' need for further support in each domain was collaboratively determined through discussion by the client and case-manager. Other outcomes of interest extracted from ASI responses included: AOD use; contraception use and subsequent pregnancies; connection to services; status of the target child; ratings of need for further help and services; and children removed into government care or returned to client care.

**2.4.3 Modifications to the ASI.** After initial baseline PCAP modified ASI surveys were performed by the case-managers, adaptations were made based on their recommendations to ensure locally relevant options were available including, but not limited to: changing measures from imperial to metric; prostitute to sex worker; adding home duties to usual employment pattern; adding methamphetamines to AOD use options; adding further types of domestic abuse; and changing the word "problems" to "issues" or "difficulties" throughout. The final modified online Australian PCAP version of the ASI is available by request as an online Qualtrics survey.

**2.4.4. Treatment Perception Questionnaire (TPQ).** The TPQ is a 10-item survey measuring satisfaction with a service and includes a single final open-ended question asking for more general comments. Each question asks for a rating on a 5-point Likert scale from strongly agree (0) to strongly disagree (4) for a total possible score of 40 points. It has demonstrated internal and test-retest reliability and has construct and discriminant validity [35]. The final question of the TPQ was an open-ended question asking clients to share any comments they had about their treatment including any areas they thought could be improved. Responses were recorded as closely as possible in real-time by MS.

## 2.5 Treatment program

All clients were enrolled in the manualised three-year case-management PCAP program modified for local Australian conditions and called "Nurturing Families." The program is based on the concepts of motivational interviewing and harm reduction and was delivered via outreach by experienced clinician case-managers. Case-managers had scheduled face to face contact with clients for at least one hour every second week, and if no other meetings were scheduled, phone calls every alternate week. Additional meetings were scheduled where necessary to facilitate service access or deal with emergencies. Case-managers assisted clients to determine their goals using the "Difference Game" card set [38]. Based on these goals case-managers help clients connect effectively with existing services, act as advocates when dealing with officials such as working with the Department of Child Protection to assist families to be re-united, provide AOD counselling and a range of other support activities. A more comprehensive overview is available at the PCAP website (https://depts.washington.edu/pcapuw/). Case-managers met with the Nurturing Families Manager at least an hour twice a month for debriefing and ensuring reliable program implementation.

## 2.6 Case-manager experience and training

The primary case-manager delivering the program was qualified in psychotherapy (Masters) with a relevant degree and had 10 years of experience working with AOD clients. A second case-manager that started four months before data collection for this study completed had a

relevant bachelor degree and 10 years of experience working in outreach and mental health settings. Six clinical psychology masters students and five assistant social-work trained case-managers with experience in AOD assessment and counselling received regular supervision and on-site support from the case-managers and site manager. These additional case-managers occasionally replaced the primary when they were unavailable. Case-managers had regular clinical supervision and training to support their professional development. Additional training was undertaken in Strong Spirit Strong Mind for working with Aboriginal clients [39], parenting training via Circle of Security and domestic violence training. The site manager was a trained psychologist with 17 years of experience in outreach, prison, mental health, hospital, child therapy, and private practice settings. The case-managers did not receive the free in-person official PCAP training program available in the US because of budgetary restraints. They were required to: read and understand the freely available online PCAP manuals, discuss the implementation of the program with the program manager, and watch two unofficial training videos available on YouTube. Consistent liaison with the original PCAP team was maintained. Therefore, the manualised intervention methodology was followed carefully while making adaptations that were considered necessary for success in the local context including further development of safety and privacy protocols.

## 2.7 Statistical analysis

**2.7.1 Missing data.** Data were missing where recall over longer periods was difficult, when the client chose not to answer questions, or when they were obviously distressed by the question. Where data were missing for variables associated with the calculated composite scores they were obtained from the case-manager during the final data collection period. Scores obtained in this way represented 31% of the total at baseline, with 3% still missing after replacing missing values from case-manager notes, and 9% at follow-up with no missing values after replacing missing values.

No survey data were collected for one client due to experiencing domestic violence issues during the follow-up data collection. One client did not complete the exit survey and TPQ due to outside interruptions during data collection.

**2.7.2 Calculation of composite scores.** Composite scores were calculated from ASI responses following the original PCAP methodology as closely as possible [28,30]. Total composite scores were calculated by summing scores based on answers to questions across five domains: 1. Utilisation of Alcohol/Drug Treatment; 2. Abstinence from Alcohol/Drugs; 3. Family Planning; 4. Health and Well-being of Target Child; and 5. Appropriate Connection with Community Services (Table 1). Each domain consisted of three to eight individual items generally scored on a five-point Likert scale (-2 indicating the most negative outcome, 0 neutral, and +2 the most positive outcome)[28]. Total possible composite total scores were different at baseline (-36–36) and follow-up (-41–45).

**2.7.3 Principal comparison.** Eight clients completed the modified PCAP ASI interviews at baseline and follow-up after approximately one year (mean = 11.3. range = 617 months). A repeated measures t-test was used to compare total scores calculated from ASI surveys at baseline and follow-up in SPSS version 25 [40]. To account for the different possible range of composite total scores at baseline and follow-up a second analysis was performed to compare differences between the percentage of total possible score at each time-point (0–100%). The number of domains for which individual clients experienced improvements was counted.

**2.7.4 TPQ qualitative content analysis.** A qualitative content analysis of the open-ended question for the TPQ was undertaken independently by MS and RW with results reached through discussion and consensus.

**Table 1. ASI composite scoring.**

| Baseline Assessment Variables | Min | Max | Endpoint Assessment | Min | Max |
|---|---|---|---|---|---|
| **Total Score Possible** | **-36** | **36** | **Total Score** | **-41** | **45** |
| **I. Alcohol/Drug TX prior to enrolment** | **-6** | **6** | **I. Alcohol/drug treatment during enrolment** | **-8** | **8** |
| Any prior alcohol/drug treatment | -2 | 2 | Any alcohol/drug treatment since enrolment | -2 | 2 |
| In/Outpatient treatment during pregnancy | -2 | 2 | Inpatient treatment since enrolment | -2 | 2 |
| Any other treatment follow-up during pregnancy | -2 | 2 | Outpatient treatment since enrolment | -2 | 2 |
| | | | Ongoing support: involved in AA, self-help groups, or lived in transitional drug-free housing | -2 | 2 |
| **II. Abstinence from Alcohol/Drugs** | **-6** | **6** | **II. Abstinence from alcohol/drugs** | **-4** | **6** |
| Abstinence during entire pregnancy | -2 | 2 | Abstinent at 36 months | -2 | 2 |
| Abstinence during 1st Trimester | -2 | 2 | Ever a period of abstinence (>1 month) since enrolment | -2 | 2 |
| Abstinence during 2nd,3rd Trimester | -2 | 2 | Long-term abstinence at 4/6 months since enrolment | 0 | 2 |
| **III. Family Planning** | **-6** | **6** | **III. Family planning** | **-8** | **8** |
| Usual pattern of birth control before conception | -2 | 2 | Usual pattern of birth control use at endpoint | -2 | 2 |
| Type of birth control method used | -2 | 2 | Type of birth control method used at endpoint | -2 | 2 |
| Target pregnancy planned | -2 | 2 | Subsequent pregnancies | -2 | 2 |
| | | | Action taken re: subsequent pregnancy (e.g., termination, alcohol/drug tx, etc.) | -2 | 2 |
| **IV. Health & well-being of target child** | **-6** | **6** | **IV. Health & well-being of target child** | **-8** | **8** |
| Adequacy of prenatal care | -2 | 2 | Target child has a regular doctor | -2 | 2 |
| Gestational age | -2 | 2 | Target child has child health nurse visits | -2 | 2 |
| Mother's ability to care for target child based on custody of previous biological children | -2 | 2 | Target child immunisation status | -2 | 2 |
| | | | Custody of target child in relation to mother's alcohol/drug use | -2 | 2 |
| **V. Family connection with services during pregnancy** | **-12** | **12** | **V. Family connection with services at follow-up** | **-13** | **15** |
| Regular family healthcare provider | -2 | 2 | Regular family healthcare provider | -2 | 2 |
| Other healthcare services (physical therapist, eye doctor, dentist) | -2 | 2 | Other healthcare services (physical therapist, eye doctor, dentist) | 0 | 1 |
| Mental health counselling | -2 | 2 | Mental health counselling | -2 | 2 |
| Childbirth/parenting classes | -2 | 2 | Assessment for alcohol/drug treatment | -2 | 2 |
| School, training classes | -2 | 2 | Parenting classes | -2 | 2 |
| Basic needs (food, clothing, supplies) | -2 | 2 | Childcare/daycare (for all children) | -2 | 2 |
| | | | Basic needs (food, clothing, supplies) | -2 | 2 |
| | | | Further education and vocational training for mothers | -1 | 2 |

# 3 Results

Time to complete the surveys ranged from 35–116 minutes (mean = 69 minutes) at baseline and from 44–98 minutes (mean = 76 minutes) at follow-up. Case-manager contact time for each client averaged 4 hours 15 minutes per month (range 3h 5m-6h 30m) with an average of 77% of attended contact hours provided by the primary case-manager (range 60%-97%), 7% by the secondary case-manager (range 1%-15%) and 17% by assistant case-managers (range 0%-29%).

## 3.1 Previous disadvantage experienced by clients

Three clients had a valid driver's license and four indicated transport was usually a problem for them at baseline. Using the 2018 poverty cut-offs from the Australian Council of Social Services (https://www.acoss.org.au/poverty/) of $433 for a single adult living alone and $909 a week for a couple with two children, six clients reported living in poverty, one was close to the cut-off, and one incarcerated. At baseline, five clients had been unemployed or busy with

home duties over the last three years, and two clients reported regular part-time work. Seven clients had been arrested and charged in their life resulting in six having convictions and three being incarcerated. Six clients had any family history of psychological difficulties. Seven reported ever being assaulted by a partner and five had been beaten during a pregnancy. Five had previous Child Protection and Family Services (CPFS) cases involving their children and one did not (two missing). Responses to the ACE survey showed that four clients reported a family history of significant alcohol problems and four a family history of significant drug problems. Of seven respondents, two had a total ACE score of two or less, and five had scores of six or more placing them in the very high-risk category.

## 3.2 Principal comparison: Composite scores from baseline to follow-up

Average composite scores improved across all domains from baseline to follow-up with the largest changes in abstinence from alcohol/drugs, family planning and connection with services (**Table 2**). There was an increase ($t(7) = 8.95$, $p < .001$) in total scores from baseline ($M = -10.4$, $SD = 4.4$) to follow-up ($M = 25.1$, $SD = 9.3$) indicating that the life situation of the clients improved. After scores were changed to proportions to account for the different possible total scores at baseline and follow-up, a similar increase ($t(7) = 8.55$, $p = < .001$) was found in total scores from baseline ($M = 0.36$, $SD = 0.06$) to follow-up ($M = 0.77$, $SD = 0.11$). Individually, one client reported improvements in three domains, three clients in four domains, and four clients in all five domains.

## 3.3 Substance use

The primary drug of concern for clients was alcohol (n = 3), cannabis (n = 2), and methamphetamine (n = 3). Reported years of alcohol/drug use ranged from 10–21 with a mean of 15 years ($SD = 3.4$). Age of first use ranged from 10–21 with a mean of 13.9 years ($SD = 3.3$). The last period of voluntary abstinence averaged 13 months ($SD = 24.2$). Three clients were abstinent at program entry with five reporting abstinence at follow-up. Use was reported by clients for the past 30 days at the baseline (alcohol & cannabis = 1, methamphetamine & cannabis = 2, cannabis = 2) and follow-up (alcohol & cannabis = 1, methamphetamine & cannabis = 1, cannabis = 1). All clients reported a decline in the volume of substance use (confirmed by the case-manager) although exact quantities were not recorded. During the year of intervention, six clients maintained AOD abstinence for over four months. The average length of time reported with no alcohol and drug use concurrently was 4.3 months ($SD = 3.3$). These changes are represented by the third largest increase in composite scores (**Table 2**).

**Table 2. ASI Scale Scores by Category, Baseline and at Follow-up.**

| Domain | Baseline | | | Follow-up | | | |
|---|---|---|---|---|---|---|---|
| | **Min** | **Max** | **Avg** | **Min** | **Max** | **Avg** | **Chg** |
| I. Alcohol/drug treatment | -6 | 6 | -0.5 | -2 | 8 | 2.3 | 2.8 |
| II. Abstinence from alcohol/drugs | -6 | -2 | -5 | 0 | 6 | 3.9 | 8.9 |
| III. Family planning | -6 | 2 | -4 | 0 | 8 | 5.9 | 9.9 |
| IV. Health & well-being of target child | -4 | 3 | 0.6 | 2 | 8 | 5.5 | 4.9 |
| V. Family connection with services | -5 | 4 | -1.5 | 2 | 14 | 7.6 | 9.1 |
| Total | -16 | -3 | -10.4 | 12 | 39 | 25.1 | |
| Change from baseline | | | | 15 | 50 | 35.5 | |

Min: Minimum score achieved, Max: Maximum score achieved, Avg: Average score across all clients, Chg = Change in average scores.

Domain scores changed the least for attendance at AOD treatment services. Six of the eight clients had attended counselling at some point before entering the program and five attended counselling during pregnancy. At follow-up all clients had received services, with two attending additional inpatient services, four additional outpatient, five additional support services which include alcohol anonymous or self-help groups, with the remaining three expressing no need for additional treatment services.

### 3.4 Family planning, subsequent pregnancies, risk of AOD exposed pregnancy

Two clients were using birth control at baseline. At follow-up, four clients were regularly using long-term birth control considered effective by WHO standards, one using less effective control, and one reported sporadic use. There was one subsequent pregnancy that resulted in a termination, therefore there were no subsequent births and no clients were currently pregnant. This domain had the largest improvement in scores across time (**Table 2**).

### 3.5 Connection to services

The majority of clients had improved connection to services at follow-up, represented by the second highest increase in composite scores for this domain (**Table 2**). Clients indicated connection to an average of five (SD = 2.7) services at baseline and eight (SD = 2.8) services after the first year. The number of additional services needed per client reduced from an average of 4.4 (SD = 3.0) to 2.3 (SD = 1.9) at follow-up. Where reported, the strength of the connection to services improved with the proportion of ratings of good (45% to 87%), acceptable (24% to 9.1%) and poor (31% to 3.6%) at baseline (n = 35) and follow-up (n = 59) respectively. All clients received AOD treatment from the case-manager and five received additional treatment from other services. At intake two women were accessing domestic violence services and at follow-up four were currently accessing services with one in extreme need. Measuring service use for the eight target children at follow-up, seven had a regular doctor, six had regular and one had non-regular health checks for their child, and seven children were fully immunised. Four mothers took a parenting class, two of which were government mandated.

### 3.6 Target child and children in care

Three target children had been born at program entry. Five women were pregnant with one in the 2$^{nd}$ Trimester (18 weeks) and four in the 3$^{rd}$ Trimester (27, 28, 34 and 35 weeks of gestation). All five target pregnancies resulted in live births with an average gestation of 37.7 weeks (SD = 1.2). Six of eight (two missing) mothers reported seeing a doctor for prenatal care and all six pregnancies were unplanned. Two target children were exposed to alcohol and four to other drugs in the 1$^{st}$ Trimester. One was exposed to alcohol and three to other drugs in the 2$^{nd}$ or 3$^{rd}$ Trimesters. Therefore, all six pregnancies reported were AOD exposed. However, all exposures were before or close to enrolment and not considered preventable by the program. At follow-up, six clients reported having a CPFS case involving the target child since enrolment. Five of the target children were living with their mother, two in foster care (CPFS) and one with their father or other relative. Target children spent an average of 6.6 months with their mother (range 0–11 months) representing an average of 54% of time in treatment (range 0%—100%).

At follow-up six clients had a child removed from their care since enrolment. At follow-up four clients had a child returned to their care. Four clients had multiple children living with them with a total of 10 children in their care. A total additional 17 children remained in state care or with another parent or relative. Fathers of target children were reported to have

difficulties with alcohol (two clients), difficulties with drugs (six clients), and psychological issues (two clients). Four fathers were involved with the target child and three were "involved to any degree" indicating less involvement but at least some contact (one missing).

### 3.7 Areas causing clients trouble and need for additional treatment

For questions assessing if clients were troubled in each domain (Fig 1), the domains they considered important for treatment (Fig 2) and the rating of need for additional support (Fig 3), there were typically two missing values for most baseline scores and no missing values at follow-up.

Only one Employment "need" score was available at baseline and three at follow-up, as this question was not asked when clients were not available for employment due to home duty responsibilities. Mean scores typically decreased across domains from baseline to follow-up (Table 3). However, mean scores increased across all three questions in the family problems domain. On average clients were more troubled by psychological issues and rated them important for therapy, however the need in this domain was generally being met. The need for help with social problems increased over time (from two of six at baseline to five of eight at follow-up). However, the question at follow-up specifically referred to the need for additional domestic violence support, with one client needing urgent support. The mean scores for employment also increased due to low numbers of responses and increased wish to participate in employment.

Comparing across domains, clients were initially most troubled by legal, mental and physical health, and drug-use difficulties. At follow-up, they were more troubled by family and

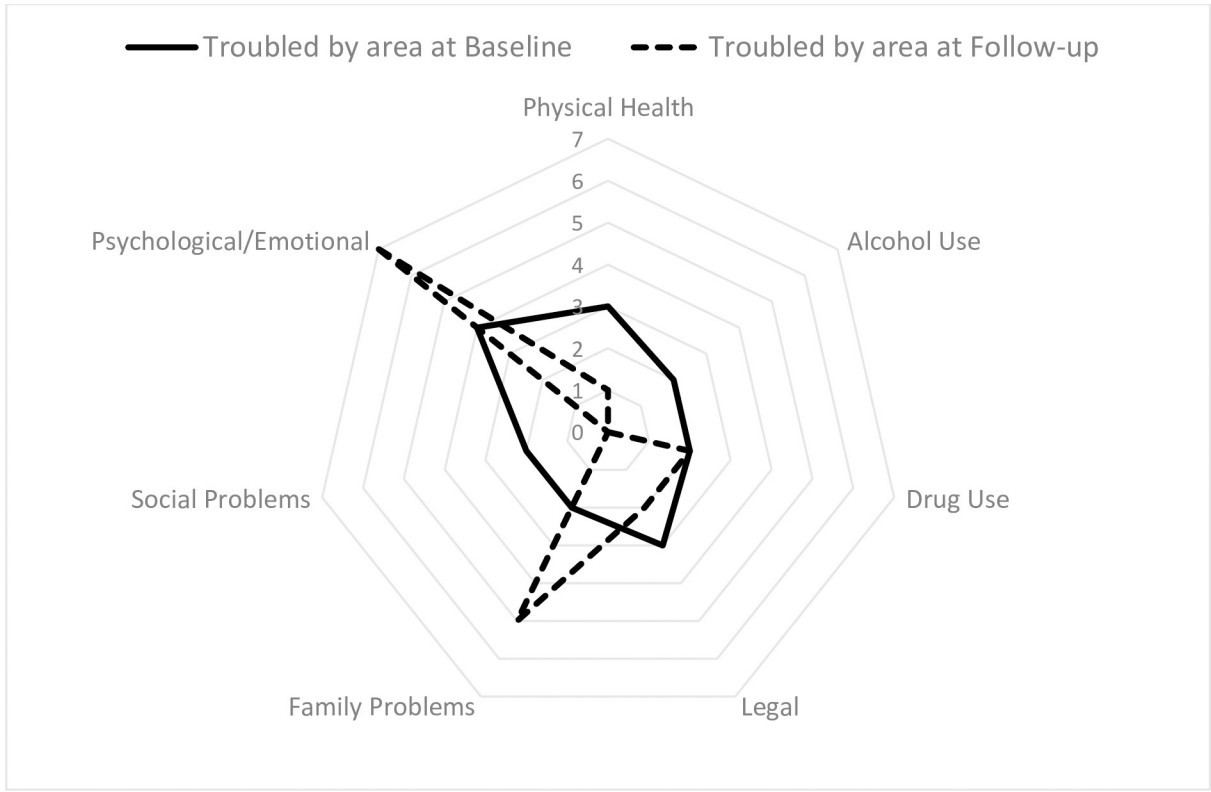

**Fig 1. Number of clients reporting being "Troubled by" each ASI category at baseline and follow-up.**

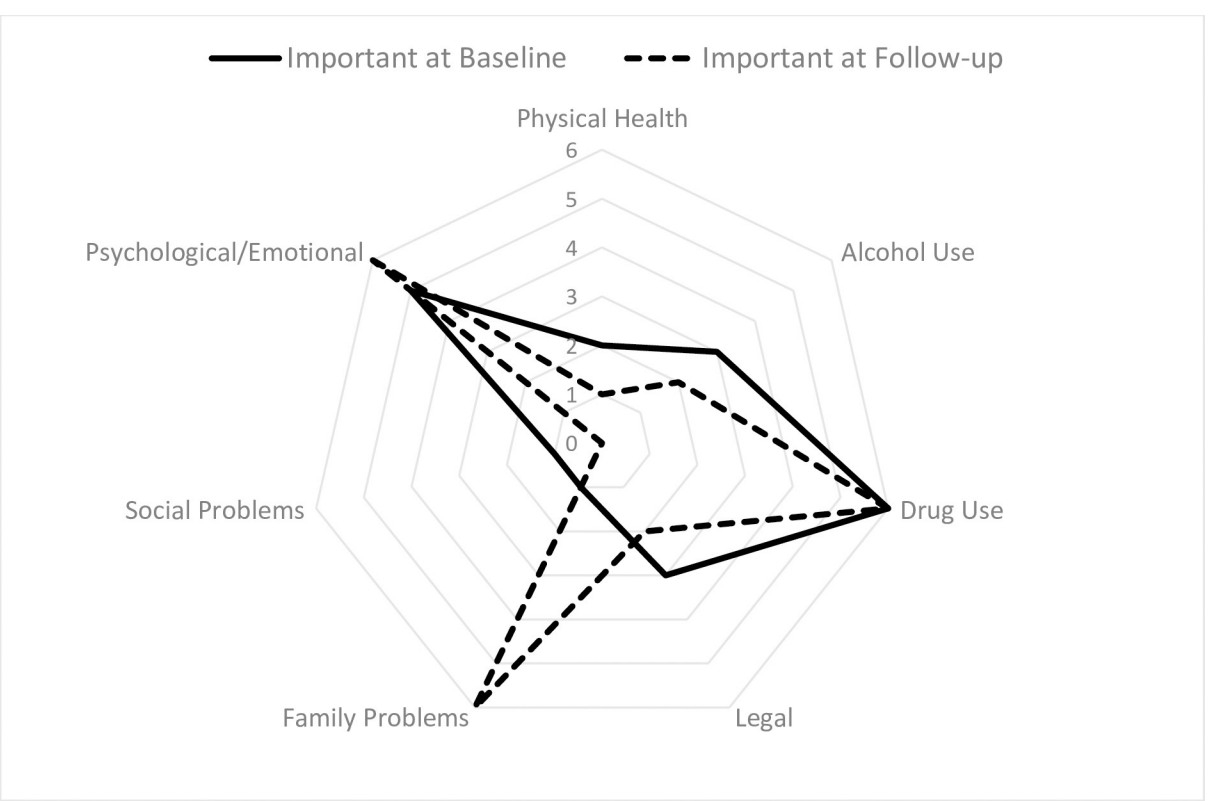

**Fig 2. Number of clients reporting each ASI category as being "Important" at baseline and follow-up.**

psychological and legal issues. Key domains reported at baseline as important for treatment were drug-use, psychological issues, and alcohol-use. At follow-up, an additional need for treatment for family problems was identified with psychological issues and drug-use remaining important for additional treatment. However, it was agreed between the case-manager and client that the key need for support initially was family planning, psychological issues and drug-use and these key needs were similar at follow-up. The total numbers of days troubled by problems over the last month at follow-up showed the issues experienced most recently were psychological, drug-use and domestic violence issues.

### 3.8 Need for additional services

The number of clients who needed more help than they were currently receiving decreased from baseline to follow-up across the following domains (with numbers in parenthesis indicating number of clients at baseline/follow-up): physical health (2/0), alcohol use (1/0), drug use (2/1), psychological difficulties (4/3) and family planning (4/1). There was still a clear need for more support around psychological difficulties. Legal difficulties remained steady (1/1) and for family difficulties (0/2) and social difficulties (0/1) the number of clients assessed as needing more addition help increased.

### 3.9 Other ASI outcomes

The majority of clients reported housing as a difficulty. There was also need for more legal services than clients were able to access. At follow-up, two clients had engaged with part-time work, one could work but hadn't found employment in the last 30 days, one was incarcerated,

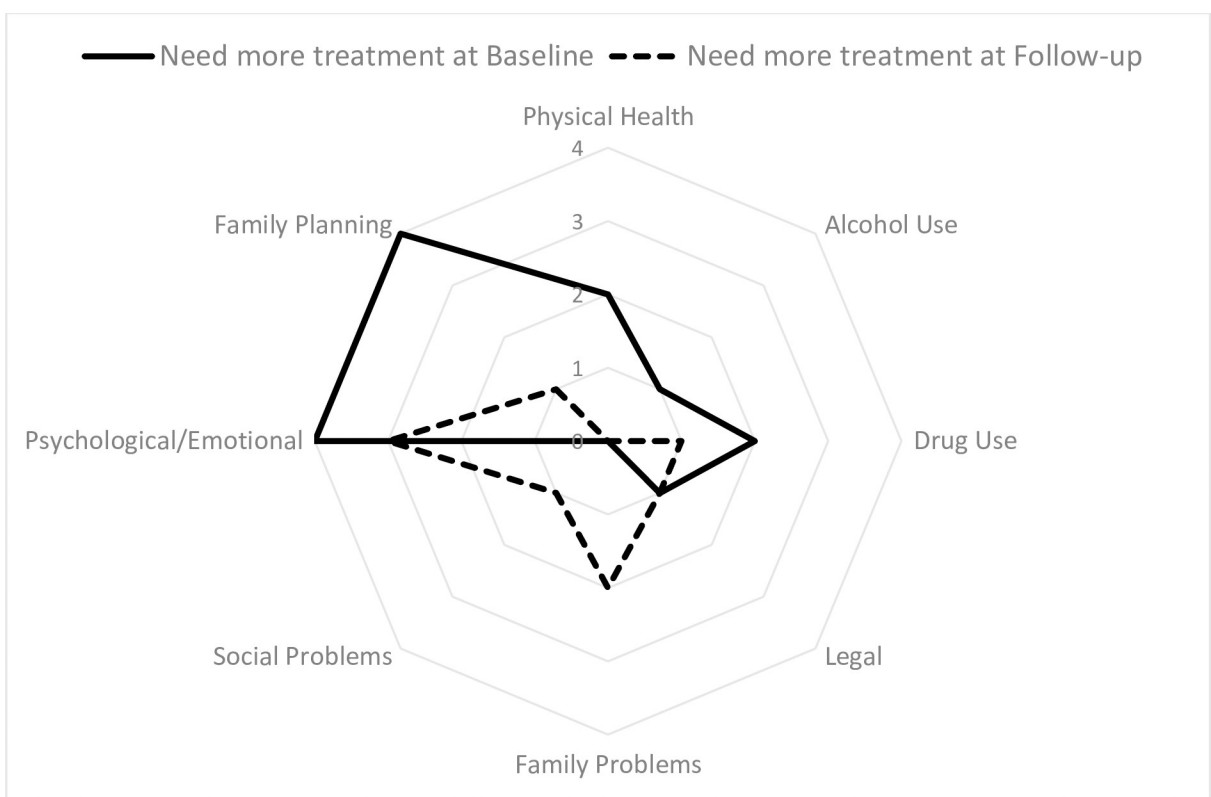

**Fig 3. Number of clients reporting the "Need for more treatment" for each ASI category at baseline and follow-up.**

and the remaining four were not able to work due to being busy with home duties. Two clients went off public assistance and one had never been on public assistance. Two clients were charged and convicted of crimes, one was on probation and one incarcerated. Five reported experiencing any type of abuse during the year.

Clients reported psychological issues in the past 30 days (number of clients at baseline/follow-up) including serious depression (5/4), serious anxiety (4/5), and having trouble

**Table 3. Means of Scores for Client Ratings of Troubled By, Number of Days Troubled in the Last Month, and Importance, at Baseline and Follow-up for ASI Categories.**

| Category | Troubled By | | Days Troubled | | Important | | Need | |
|---|---|---|---|---|---|---|---|---|
| | **Bl** | **FU** | **Bl** | **FU** | **Bl** | **FU** | **Bl** | **FU** |
| Physical Health | 1.0 | 0.3 | 21 | 15 | 1.3 | 0.3 | 1.0 | 0.1 |
| Employment | 0.0 | **1.3** | 0 | 16 | 0.0 | **0.7** | 0.7 | **1.0** |
| Alcohol Use | 0.5 | 0.0 | 7 | 6 | 1.7 | 1.0 | 0.8 | 0.3 |
| Drug Use | 1.0 | 0.9 | 78 | 37 | 3.3 | 2.5 | 1.2 | 0.9 |
| Legal | 1.3 | 1.0 | | | 1.0 | 1.0 | 0.7 | 0.6 |
| Family Problems | 0.7 | **2.0** | 3 | 7 | 0.6 | **2.3** | 0.3 | **1.4** |
| Social Problems | 0.7 | 0.0 | 2 | **32** | 0.2 | 0.0 | 0.3* | **0.9**\* |
| Psychological/Emotional | 1.7 | **2.8** | 55 | **114** | 2.3 | **2.6** | 1.7 | 1.3 |
| Family Planning | n/a | n/a | | | n/a | n/a | 2.0 | 0.3 |

**Bl**: Baseline; **FU**: Follow-up. Maximum score is 3. Target child had no need rating in the follow-up survey. The bolded cells indicate scores that increased from baseline to follow-up. *The "Need" column for Social Problems is particularly for help with domestic violence.

concentrating or understanding at some point (5/4). At follow-up four were prescribed medication for psychological or emotional difficulties in the past 30 days. At baseline 57% of clients (one missing) reported experiencing at least one day of psychological or emotional issues in the past 30 days which increased to 88% at follow-up. Similarly, the average number of days experiencing issues increased from an average of 13.8 at baseline to 14.3 days at follow-up.

### 3.10 Satisfaction

Seven of eight participants provided TPQ responses, with one participant not completing due to time restrictions. Six participants reported the maximum possible TPQ satisfaction score (40 out of 40), with the seventh participant reporting a score of 35 (mean = 39.3).

Out of the seven clients who responded to the final qualitative question of the TPQ, five directly expressed they were happy with the program or directly recommended it with comments such as: "Awesome program, recommend to anyone" and "Loved the service." Five clients directly said that there was nothing to improve. Four said it had been life-changing, for example, "Has changed my life, complete different person to when I started." Three clients noted how important it was that the help provided was "supportive", "understanding", "friendly and non-judgmental", and that they could tell the case-manager anything. Four noted directly how friendly the case-manager was. Six of the seven noted how helpful the program was such as: ". . .above and beyond helpful. . .", "helped me control my emotions", "helped me reach my goals", and "helped make small connections with children positive." The comprehensiveness of support was noted by three clients: "Can get extra help when needed", "Really liked home visits otherwise hard without transport," "Also supported partner," "Anything out of scope can seek and find additional help."

The suggestions for improvement provided by clients included recommendations that government provide more funding for this service (one client), and to improve the communications and integration between Nurturing Families and CPFS (one client).

## 4 Discussion

This paper reports a pilot evaluation of the PCAP program delivered in Australia. Overall, the results provide evidence of significant improvement in the lives of clients at follow-up, approximately one year post baseline. The total composite scores calculated for this study were within one point of similar scores calculated at program entry ($M$ = -11.0) and after three years ($M$ = 26.0) for PCAP replication studies in the US [30]. Individually, clients reported improvement in at least three of the five domains measured, with five clients improving across all five domains. The key goals of the PCAP program were fulfilled: 1) There were reductions in AOD use, 2) There was increased use of contraception, and 3) The number of services accessed increased. It should be noted that the results reported here are one-year outcomes for this three-year program. For women with complex needs and vulnerabilities, recovery is not typically linear, so continued support is essential to meet future challenges they are likely to face, including relapse. Over the longer-term, we anticipate that ongoing engagement in the program and connected services will consolidate these improvements. There is currently a real unmet need with a long waiting list for the program.

Alcohol and drug use decreased for all clients, with 75% of clients achieving periods of abstinence of at least four months. Alcohol use was not as prevalent as expected, with more clients facing difficulties with methamphetamines and cannabis. This may be reflective of a change in the underlying drug use patterns in Australia [1]. While overall drug use was an important cause of problems and the need for treatment, other issues, such as psychological and physical health, were equally or more important to clients. This is the first time the results

for these sections of the ASI have been reported for the PCAP program and provide insight into the full range of needs that clients with problematic AOD use experience.

Analysis of specific domains demonstrated that the largest improvement was found in the use of family planning, with nearly all clients using effective birth control at follow-up. This is an important aspect of reducing the number of children born exposed to alcohol and drugs which is often poorly evaluated in FASD prevention programs [41]. For some clients, the use of effective birth control, especially in the short-term, may be an easier solution than complete abstinence from alcohol and drug use.

The characteristics of women who took part in our study and their outcomes in terms of reducing risk of an AOD exposed pregnancy were similar to other studies where the PCAP has been trialled in the US [30] and Canada [27]. To our knowledge this was the first time changes in areas in which clients expressed concern and need for further treatment was reported for PCAP. Similar programs are rare or not often evaluated. A recent systematic review of programs for prevention of prenatal alcohol exposure and FASD identified PCAP as the only long-term support program found effective in evaluations with four other programs showing some short-term reductions that were not sustained [42]. Short-term brief interventions were the most popular approach and while they showed some promise for reducing alcohol exposed pregnancies only four recruited from women at higher risk of PAE and only one had a follow-up period of 12 months with the other 14 having follow-ups of 6 months or less, making long-term benefits uncertain [42]. No other studies found were conducted in Australia [42]. As prevention of alcohol use during pregnancy has unique opportunities and challenges, it is not appropriate to make comparisons with approaches designed to reduce general alcohol use.

All clients received AOD treatment during the program. AOD treatment service access scores increased least across time, which may have represented a true change or been a product of the composite scoring system. Additional points were awarded for attending different types of treatment (Inpatient, Outpatient, and support groups). Given case-managers provided more direct AOD counselling to clients than in typical PCAP programs, clients may have felt less need to seek additional outside services.

The fact that psychological issues were rated as troubling and requiring extra treatment was not surprising given at least half of clients reported depression, anxiety, and cognitive difficulties in the past 30 days at both time-points. The questions about psychological issues were prefaced with, "Have you had a significant period (that was not a direct result of drug/alcohol use) in which you have experienced:". While this does not guarantee the issues were not related to AOD use, it points to clients experiencing difficulties in these domains for other reasons. There were corresponding increases in the importance of addressing family problems and social problems which could be explained by assessment timing. These issues may have flared up spontaneously, or clients may have been addressing their family and social issues in more depth having met basic physiological and safety-security needs, following predictions based on Maslow's Hierarchy of Needs [43]. The number of clients accessing services for mental health also increased over time, indicating that clients were actively engaged in meeting these additional needs.

The clients valued the program highly, with many highlighting the number of ways that it had assisted them to make positive changes in their lives. They all praised the professionalism and care of the case-manager delivering the program. Scores on the TPQ indicated that all participants reported a high degree of satisfaction with the program. The mean score of 39.3 was higher than other studies of satisfaction across a range of alcohol treatment services of 27.7 [35], 25.1 [44] and 20.08 [45]. The level of satisfaction with substance-use treatment can predict employment, mental health and financial outcomes [45] and is therefore an important metric against which to assess program potential.

There were differences in program implementation compared to the manualised PCAP program. A major difference was that the case-managers had not undergone the PCAP training, but only read the comprehensive online PCAP manual and watched training videos. This may have been offset by the case-managers having more qualifications and experience in providing drug and alcohol therapy than typical PCAP advocates. Other differences included the case-manager providing more direct AOD counselling and having less supervision from the program-manager. Given the significant positive changes made by the women in the program, these modifications don't appear to have impacted program delivery negatively. Having case-managers present at the follow-up interviews provided two benefits. Firstly, they helped the clients who re-experienced trauma symptoms when answering questions during the survey. Secondly, they helped clients recall additional instances of AOD use and services accessed which helped improve accuracy of data collected given the long timeframe used for some questions.

There were contextual issues that may have affected experiences of the program for some clients including experiences of domestic violence and availability of services. For those clients currently needing services for domestic violence (5 out of 8), depending on the type and behaviour by the partner, may have limited the interactions with the case-manager both in terms of time spent together and the topics of discussion when meeting. Also, clients at times reported their interactions with various services available as poor. If there were no practical alternatives available locally they may not have been able to access services with good fit which may have led to reduced therapeutic benefit. At times good services clients were attending lost funding and were closed. Further examination of these contextual issues, case-manager perceptions of the program and the full suite of implementation details will be undertaken after the first cohort of clients finish the program by applying the Consolidated Framework for Implementation Research (CFIR).

There were limitations in this pilot study. There was a significant proportion of data missing at the baseline survey before these were replaced by the case manager providing these details directly to the researcher. This was primarily due to clients refusing to answer questions, sometimes entire sections, at baseline. As with all research, participation was voluntary and clients could refuse to answer questions. The baseline interviews were conducted at the first meeting, and a lack of rapport with the case manager may have contributed to the volume of missing data. This was not anticipated, as we followed the protocols of published PCAP research and this had not previously been reported as a problem. However, missing data were provided to the case manager later in treatment, indicating question timing is important to client comfort.

Another limitation was the accurate assessment and tracking of AOD use over time. The ASI 5th edition concentrates on assessing frequency of use rather than quantity, using reporting in the last 30 days. While it has been argued this is a reliable approach [34], clients in the present study often found this difficult. Therefore, exact amounts were not estimated and exact reporting of reductions in volume were not possible. At baseline, clients reported that assessing lifetime use was difficult and they may have been reluctant to report all use before clinical rapport had been firmly established. The PCAP modified ASI follow-up questions about AOD use *since enrolment* were also often difficult for clients to recall accurately. The absolute time at risk of AOD-exposed pregnancy could not be calculated as the overlap between AOD use and effective contraceptive use was not available from the ASI. These issues could potentially be resolved with more frequent future measurement of AOD use and contraception.

Another potential issue arose from having case-managers involved with collecting baseline data and attending the follow-up interview. The ASI is an effective clinical instrument and we attempted to integrate it into normal clinical practice for baseline measurements. Efforts were

made to reduce the impact of social desirability on client responses by make it clear during consent that their treatment would not be affected in any way by participation in the research study, or non-participation. It is also a key pillar of PCAP that clients are not refused treatment for revealing AOD, crime, or other personal details with case-managers adopting a non-judgemental attitude. To some extent, we believe that the trust between the case managers and the participants increased the likelihood of disclosure, thereby increasing the reliability of the findings. Despite this, the effects of social desirability may have biased client responses. In the future adopting a new approach from the US where supervised and ASI trained psychology Masters students conduct all interviews may be more appropriate.

Existing community services were inadequate to meet the needs of these clients in some areas. Governments could address this by more accurately assessing community need for services and appropriately budgeting for essential, good quality, community services. For the purposes of service planning and budgeting it is important to understand the proportion of the population experiencing similar complex needs. As the PCAP program is based on connecting clients with existing services, if services are not readily available program efficacy may be compromised. The continuation of the service is based on further funding which is currently unstable and unable to meet the needs of the growing number of women applying to the program. Also, due to the intense work by the case managers it is important to keep case-loads manageable to avoid burnout, therefore funding for more case managers is important. Finally, funding for key staff to attend the free training available in the US would allow them to improve their skills and train other current and future staff which could lead to further improvements for clients.

## 4.1 Future work

Problems with relying on the ASI for evaluation have been identified by other researchers in Canada [46]. Therefore, while maintaining compatibility with the old measurement approach by retaining a minimal set of key ASI questions, the evaluation will be updated based on recent research into a common framework for evaluating FASD prevention programs [47] now being used for a multi-site evaluation.

A broader range of more relevant evaluation measures for clients and case-managers could be accessed through a process of co-design. Involving consumers in all areas of health research is considered an important principle of good research practice and guidelines for critical appraisal of the quality and impact of consumer involvement are available [48]. Participatory evaluation attempts to involve all stakeholders with an interest in the outcomes by making changes based on their recommendations [49]. Participatory research allows marginalized groups to feel empowered, and can facilitate engagement, equity, equality and improved outcomes for research for participants [50,51]. More frequent measurement of alcohol, drug and contraceptive use will be explored in future evaluation, although administrative burden on the case-managers is an important issue. The timing of asking sensitive questions at the start of treatment will also be addressed.

Given many PCAP clients experience mental health issues researchers in the US have hypothesised that the program may be strengthened by providing case-managers with further training in this area. Therefore, they have conducted a promising trial addition of an adapted version of the culturally relevant MomCare mental health program [52] which appeared to reduced depression in clients and applied for further funding for a full evaluation. If further results are promising the program will be considered for the Australian Nurturing Families PCAP in Australia.

It is currently not known if the PCAP program will differ in effectiveness for those who are younger or have had difficulties with AOD over shorter periods and those who are older or have longer AOD histories. This is an important question to consider as in Australia women with moderate and high alcohol use during pregnancy were more likely to have first become intoxicated before the legal age of 18 and high sustained drinking more common in women over 35 [53].

As this is a pilot study, a larger trial, potentially across more sites, would be appropriate.

## 5 Conclusions

The findings in this pilot study were promising with overall significant improvements in client lives indicated by ASI composite rating scores, reduced AOD use, no further exposed pregnancies, and all but one client giving the highest possible satisfaction scores. Important lessons were learnt about the evaluation. In the future, more effective measures of all areas of progress are needed. Different evaluation tools will be considered to this end.

## Acknowledgments

We would like to acknowledge Women's Health and Family Services for providing the Nurturing Families intervention and inviting the researchers to evaluate the program. The original PCAP team at the University of Washington provided valuable advice, help and support and we thank them for sharing the program in detail with the world for free. We especially thank the clients who participated for their courage in sharing their histories and victories.

## Author Contributions

**Conceptualization:** Martyn Symons, Amy Finlay-Jones, Jennifer Meehan, Natalie Raymond, Rochelle Watkins.

**Data curation:** Martyn Symons.

**Formal analysis:** Martyn Symons.

**Funding acquisition:** Rochelle Watkins.

**Investigation:** Martyn Symons, Jennifer Meehan.

**Methodology:** Martyn Symons, Amy Finlay-Jones, Natalie Raymond, Rochelle Watkins.

**Project administration:** Martyn Symons.

**Supervision:** Rochelle Watkins.

**Writing – original draft:** Martyn Symons.

**Writing – review & editing:** Martyn Symons, Amy Finlay-Jones, Jennifer Meehan, Natalie Raymond, Rochelle Watkins.

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
