## [Decision Letter · Decision Letter 0]

9 Mar 2022

PGPH-D-21-00670

Nurturing Families: One year pilot outcomes for a modified Parent Child Assistance Program in Australia

Dear Dr. Symons

Thank you for submitting your manuscript to PLOS Global Public Health. After careful consideration, we feel that it has merit but does not fully meet PLOS Global Public Health’s publication criteria as it currently stands. Therefore, we invite you to submit a revised version of the manuscript that addresses the points raised during the review process.

We look forward to receiving your revised manuscript.

Kind regards,

Tara Carney, Ph.D.

Academic Editor

Journal Requirements:

1. Please update your Competing Interests statement. If you have no competing interests to declare, please state: “The authors have declared that no competing interests exist.”

Additional Editor Comments (if provided):

Reviewers' comments:

Reviewer's Responses to Questions

**Comments to the Author**

1. Does this manuscript meet PLOS Global Public Health’s publication criteria? Is the manuscript technically sound, and do the data support the conclusions? The manuscript must describe methodologically and ethically rigorous research with conclusions that are appropriately drawn based on the data presented.

Reviewer #1: Yes

Reviewer #2: Yes

2. Has the statistical analysis been performed appropriately and rigorously?

Reviewer #1: Yes

Reviewer #2: I don't know

3. Have the authors made all data underlying the findings in their manuscript fully available (please refer to the Data Availability Statement at the start of the manuscript PDF file)?

Reviewer #1: Yes

Reviewer #2: No

4. Is the manuscript presented in an intelligible fashion and written in standard English?

Reviewer #1: Yes

Reviewer #2: Yes

5. Review Comments to the Author

Reviewer #1: To the Authors

Thank you for the opportunity to review your paper. The paper offers valuable insights into the value of the PCAP program in an Australian context. The paper is well-written and neatly organised. I have listed some recommendations to strengthen your paper.

Page 12: Please clarify if the implementers received or did not receive the free training plan as you indicate that they did not receive the free plan but subsequently explain that they did. Later in the paper you indicate that they did not participate in formal training but had access to the free manuals. Perhaps I am misunderstanding, but please try to clarify.

P24- 4.11: It would be useful to draw up a table with some qualitative data that describe the case managers perceptions of the programme.

General questions

- Did you observe any age related differences between adult youth and middle adults

- Could you comment on any contextual issues you might have observed or be aware of that could have influenced experiences of the program?

- How do the findings compare with other contexts?

- How do the findings compare to other/similar interventions or programs aimed to collectively or individually address the issues targeted in the paper? Are there other programs or elements of other programs that could be incorporated in future work that could potentially strengthen the program or making more contextually relevant?

In all, the paper offers valuable insights on the potential feasibility of the PCAP program in an Australian context. I wish the authors well in finalizing their paper towards publication.

Thanks

Reviewer #2: The authors present a very well written paper on a well thought through and interesting study. However, the very small sample size is very concerning for a quantitative study even as the authors refer to it as a pilot study. My recommendation is that the authors provide a sample size calculation and the editor, in consultation with a statistician, gives this limitation serious consideration.

1. There is mention of Aborigines in the paper without an indication that this population is a focus of this study, which indeed from reading of the entire paper it is clear that this population is not the focus. I am concerned about this group being singled out as having a high FASD problem - perhaps the authors can also talk about other special populations where FASD prevalence is high rather than mention only one population.

2.a) The case-managers seem far too involved in the research process. They were both interventionists and researchers to a very small sample of women who were in need of resources that the case-managers were providing. All these dynamics create a huge potential for social desirability among the participants. The reason that the case-managers (rather than the researchers) conducted the baseline interviews is not currently not clear. Furthermore, the need for case managers to assist participants with recall, thus having to have them present in the interviews is also not convincing. In fact, the case-managers involvement seems to be viewed positively by the authors without reflection on how this may have biased the findings.

b) "WHFS approached the authors to evaluate a pilot PCAP program in Perth that had received funding for one case-manager." This text suggests that only one case manager was involved in the study but subsequent parts of the paper mention case-managers (plural) being involved. Some clarification here would be useful.

3. The description of the case manager interviews is too little - both in the methods and the findings. In some sections of the methods there is no mention of this component at all. At the moment it does not come across as an equally valuable part of this study and it is not well integrated with the rest of the paper. I think the paper presents too much at once, perhaps this component could be written up in a different paper. Relatedly, the design of the study does not currently meet the definition of a mixed methods study - i.e. as a third methodological approach alongside quantitative and qualitative methods. The authors would do well to reconsider use of this term for their study.

6. PLOS authors have the option to publish the peer review history of their article (what does this mean?). If published, this will include your full peer review and any attached files.

**Do you want your identity to be public for this peer review?** For information about this choice, including consent withdrawal, please see our Privacy Policy.

Reviewer #1: **Yes: **Candice Groenewald

Reviewer #2: No

---

## [Decision Letter · Decision Letter 1]

13 Jul 2022

Nurturing Families: One year pilot outcomes for a modified Parent Child Assistance Program in Australia

PGPH-D-21-00670R1

Dear Dr Symons,

We are pleased to inform you that your manuscript 'Nurturing Families: One year pilot outcomes for a modified Parent Child Assistance Program in Australia' has been provisionally accepted for publication in PLOS Global Public Health.

Best regards,

Tara Carney, Ph.D.

Academic Editor

Reviewer Comments (if any, and for reference):

Reviewer's Responses to Questions

**Comments to the Author**

1. If the authors have adequately addressed your comments raised in a previous round of review and you feel that this manuscript is now acceptable for publication, you may indicate that here to bypass the “Comments to the Author” section, enter your conflict of interest statement in the “Confidential to Editor” section, and submit your "Accept" recommendation.

Reviewer #1: All comments have been addressed

Reviewer #2: All comments have been addressed

2. Does this manuscript meet PLOS Global Public Health’s publication criteria? Is the manuscript technically sound, and do the data support the conclusions? The manuscript must describe methodologically and ethically rigorous research with conclusions that are appropriately drawn based on the data presented.

Reviewer #1: Yes

Reviewer #2: Yes

3. Has the statistical analysis been performed appropriately and rigorously?

Reviewer #1: Yes

Reviewer #2: Yes

4. Have the authors made all data underlying the findings in their manuscript fully available (please refer to the Data Availability Statement at the start of the manuscript PDF file)?

Reviewer #1: No

Reviewer #2: (No Response)

5. Is the manuscript presented in an intelligible fashion and written in standard English?

Reviewer #1: Yes

Reviewer #2: Yes

6. Review Comments to the Author

Reviewer #1: The authors have submitted a revised version of their paper which can be accepted for publication. However, the authors have requested that their data not be made publicly available with valid reasons.

Well done to the authors for their efforts thus far.

Reviewer #2: (No Response)

7. PLOS authors have the option to publish the peer review history of their article (what does this mean?). If published, this will include your full peer review and any attached files.

**Do you want your identity to be public for this peer review?** For information about this choice, including consent withdrawal, please see our Privacy Policy.

Reviewer #1: No

Reviewer #2: No
